A multitask model for realtime fish detection and segmentation based on YOLOv5

Liu QinLi
Gong Xinyao
Li Jiao
Wang Hongjie
Liu Ran
Liu Dan
Zhou Ruoran
Xie Tianyu
Fu Ruijie
Duan Xuliang duanxuliang@sicau.edu.cn
College of Information Engineering, Sichuan Agricultural University , Ya’an, Sichuan , China
Hemanth Jude
Electronic publication date: 2023 Mar 10
Publication date: 2023
Volume: 9
Electronic Location ID: e1262
Received 2022 Oct 24; Accepted 2023 Feb 1
Copyright: © 2023 Liu et al.
Copyright year: 2023
Copyright holder: Liu et al.
License: This is an open access article distributed under the terms of the Creative Commons Attribution License, which permits unrestricted use, distribution, reproduction and adaptation in any medium and for any purpose provided that it is properly attributed. For attribution, the original author(s), title, publication source (PeerJ Computer Science) and either DOI or URL of the article must be cited.
License URL: https://creativecommons.org/licenses/by/4.0/

Keywords: Realtime monitoring, Golden crucian carp dataset, Multi-task, Object detection, Semantic segmentation, YOLOv5

Funding: The authors received no funding for this work.

==============================
The accuracy of fish farming and real-time monitoring are essential to the development of “intelligent” fish farming. Although the existing instance segmentation networks (such as Maskrcnn) can detect and segment the fish, most of them are not effective in real-time monitoring. In order to improve the accuracy of fish image segmentation and promote the accurate and intelligent development of fish farming industry, this article uses YOLOv5 as the backbone network and object detection branch, combined with semantic segmentation head for real-time fish detection and segmentation. The experiments show that the object detection precision can reach 95.4% and the semantic segmentation accuracy can reach 98.5% with the algorithm structure proposed in this article, based on the golden crucian carp dataset, and 116.6 FPS can be achieved on RTX3060. On the publicly available dataset PASCAL VOC 2007, the object detection precision is 73.8%, the semantic segmentation accuracy is 84.3%, and the speed is up to 120 FPS on RTX3060.

Introduction

Humans have known how to catch fish since ancient times, and now the demand for fish is increasing year by year. The nutritional value of fish is extremely high and fish is easy to absorb, which is popular with people. In the process of the aquaculture, managers need to gain information about the lives of fish, shrimp, shellfish and other aquatic organisms, specifically, species, behavior identification and biomass estimation (total weight of fish and shrimp in specific waters). Among them, the biomass estimation is the total weight of fish and shrimps in a specific water. The information on the biomass of fish, shrimp and other aquatic organisms at various growing stages is crucial, because managers need to optimize feeding requirements and make effective decisions based on this information.

The conventional estimation of fish biomass mainly adopts manual fishing and weighing, which not only consumes human resources and decreases efficiency, but also may have adverse effects on the growth of fish. Since there is a relationship between the weight of organisms, and their body length and image area, the weight can be estimated indirectly through a deep learning-based image segmentation method to predict daily feed intake of aquatic organisms. It can also monitor the growth rate of aquatic organisms, control breeding density, determine optimal harvesting time, and ensure optimal utilization of facility investment. Due to the reliance on visual pattern matching in recognition process, it is convenient and least invasive compared to traditional manual monitoring. Non-invasive techniques offer significant advantages in terms of cost, safety and convenience. Therefore, using computer vision methods for intelligent detection of fisheries is an inevitable trend in the development of fish farming industry in modern society (Lin et al., 2021).

There are mainly four tasks in computer vision regarding image recognition: classification, location, detection, and segmentation. Classification focuses on “What broad category of object is in this photograph” (Brownlee, 2019), and is concerned with the overall content of the image. It is mainly divided into two types of tasks: dichotomous and multiclassification tasks, and is widely used in agriculture, medicine, soil, etc. (Ashraf & Khan, 2020; Liaqat et al., 2020; Srivastava, Shukla & Bansal, 2021). The main task of localization is to find where the object is, generally in the form of a bounding box to circle the location of the object (Sermanet et al., 2013). Unlike classification, detection is to find out which objects are in the photograph and obtain their category information and location information, which is a combination of classification and location (Brownlee, 2019). Segmentation mainly focuses on pixels, solving the problem of “What pixels belong to the object in the image” (Brownlee, 2019). Segmentation includes semantic segmentation and instance segmentation. The semantic segmentation is to determine whether pixels go well with the object without distinguishing different instances of the same category, while the instance segmentation needs to determine different instances of the same category on the basis of semantic segmentation, which is a combination of object detection and semantic segmentation.

Object detection is a collection of classification and regression, mainly aiming to find out all the targets of interest in an image and determine their categories and locations. Early object detection algorithms were based on handcrafted features. The representative ones are Viola-Jones detectors (Viola & Jones, 2001), HOG detectors (Dalal & Triggs, 2005) and Deformable part based model (DPM) (Felzenszwalb, McAllester & Ramanan, 2008). These algorithms have laid an important foundation for later object detection algorithms. The development of object detection algorithms was limited due to the saturation of handcrafted features. However, with the advent of convolutional neural networks, object detection begins developing unprecedentedly, and object detection algorithms based on deep learning has become mainstream. Deep learning-based object detection algorithms are mainly divided into two categories, One-Stage and Two-Stage. Common algorithms for One-Stage include OverFeat (Sermanet et al., 2013), YOLOv2 (Redmon & Farhadi, 2017), YOLOv3 (Redmon & Farhadi, 2018), YOLOv4 (Bochkovskiy, Wang & Liao, 2020), SSD (Liu et al., 2016), and RetinaNet (Lin et al., 2017), while for Two-Stage there are R-CNN (Girshick et al., 2014), SPP-Net (He et al., 2015), Fast R-CNN (Girshick, 2015), and Faster R-CNN (Ren et al., 2015), etc. Currently, these algorithms are widely used in many fields such as security, military, transportation, and daily life, etc. (Saikia et al., 2017; Janakiramaiah et al., 2021; Li et al., 2020; Jiang et al., 2019). Among the algorithms for object detection, One-Stage’s object detection algorithm enables a completely single training of shared features and rapid acceleration, while guaranteeing accuracy. For example, YOLOv1 enables real-time object detection with the basic network running at 45 frames per second and processes streaming videos in real time with a latency of less than 25 ms (Redmon et al., 2016).

As a quintessential computer vision problem, semantic segmentation involves taking some raw data (e.g., planar images) as input and converting them into masks with highlighted regions of interest. Semantic segmentation is mainly divided into standard semantic segmentation and instance-aware semantic segmentation, commonly using FCN (Long, Shelhamer & Darrell, 2015), OCRNet (Yuan et al., 2019), Deeplabv3+ (Chen et al., 2018), and UPerNet (Xiao et al., 2018). These algorithms are mainly used in self-driving cars, geological detection, facial segmentation, precision agriculture, clothing classification, and other fields (Masood et al., 2022; Hu et al., 2020; Khan, Mauro & Leonardi, 2015; Du et al., 2019; de Souza Inácio, Brilhador & Lopes, 2019).

Instance segmentation is currently a comparatively mature technique in the field of real-time scene understanding and image information processing (Xin-yu et al., 2020). It is the localization of instances in an image using a object detection algorithm, and then on the basis of semantic segmentation, the target objects in different localization frames are further segmented into specific objects of classified categories. In contrast to the bounding box of object detection, instance segmentation can be accurate to the edges of objects, while compared with semantic segmentation, it needs to annotate different individuals of the same object on the graph. The existing instance segmentation networks, such as Mask R-CNN (He et al., 2017) and Faster R-CNN (Ren et al., 2015), have the high rate of accuracy, however, they fail to process details such as segmented edges meticulously and have a low speed.

All stages of fish growth and development are susceptible to various external factors, possibly resulting in poor growth or even death of fish, which brings serious losses to fish farmers. The focus of current intelligent fishery research is the way to analyze and understand the state of fish growth process in a timely and effective manner. Therefore, our research will focus on the real-time nature of the model, so that we can later deploy it on cell phones or monitoring devices to be applied to fish farms for real-time monitoring of fish growth process. In recent years, researches on fish image detection segmentation based on deep learning have attracted great attention. Chen, Sun & Shang (2017) proposed an automatic fish classification system based on deep learning, Akgül, Çalik & Töreyın (2020) proposed fish detection in turbid underwater, Li, Tang & Gao (2017) proposed a deep and lightweight network for detecting fish, Wang et al. (2020) proposed a detection of abnormal behaviors of underwater fish using artificial intelligence techniques, Alshdaifat, Talib & Osman (2020) proposed a deep learning framework for segmentation of fish with underwater videos, Knausgård et al. (2022) proposed a method for detecting and classificating temperate fish, Parida (2019) proposed a hybrid transition region color image segmentation method based on dual transition region extraction for fish image segmentation applications. Lei, Ouyang & Xu (2018) proposed an image segmentation method based on equivalent 3D entropy and artificial fish population optimization algorithm, which is more efficient than the traditional 3D entropy method and the equivalent 3D entropy method. Raza & Song (2020) proposed a quick and accurate method for fish detection based on the improved YOLO-v3 model and migration learning. Its average mean accuracy precision (mAP) of 87.56%, which increased from 87.17% to 91.30%, compared with the experimental analysis of the YOLOv3 model and the improved model. Cai et al. (2020) proposed an improved YOLOv3 fish detection model based on MobileNetv1 as the backbone, which combined YOLOv3 with MobileNetv1 for real detection of farmed fish. Arvind et al. (2019) used the Mask R-CNN model along with the GOTURN tracking algorithm for fish detection and tracking, showing that image multi-region parallel processing and tracking are accurate with F1 score of 0.91 at 16 frames per second inland. Nevertheless, most methods mentioned above suffer from slow segmentation and low speed,and also cannot be better monitored in real time. Thus, we use a multi-task network YOLOv5 as the backbone network and object detection branch, combined with semantic segmentation head optimized by the GhostC3 module, improving the speed and accuracy of the model in general.

In short, our main contributions are: The construction of a golden crucian carp segmentation dataset: Because of few public datasets of fish segmentation, we have built a new golden crucian carp dataset, containing 640 semantic segmentation images of 10 golden crucian carps.

Providing a comparison to the existing mainstream instance segmentation networks, object detection networks and semantic segmentation networks.

Detecting and segmenting golden crucian carp based on YOLOv5. We use a multitasking model to solve the problem of monitoring fish and acquiring fish segmentation images in real time, which perform real-time detection and segmentation while ensuring high accuracy. At the same time, we used the Ghost module to further optimize the model and reduce the number of parameters of the model. Compared with most of the networks framed by instance partitioning, the FPS of our model is higher and can reach the standard of real-time monitoring.

Materials and Methods

Multi-task learning model is learning multiple tasks together, and the efficiency and quality of studying each task can be improved by learning the connections and differences between different tasks. Different from traditional single-task learning model, which seeks to use a specific model (Ma et al., 2018) to accomplish the task, multi-task learning model is more effective to improve the generalization of models. In addition, computational repetition can be minimized, inference speed can be increased, and memory utilization can be reduced by sharing layers across multiple tasks (Lu et al., 2022).

In this article, our dataset is divided into two types of annotations, and our model also includes two tasks: object detection and semantic segmentation of golden crucian carp. These two share the backbone based on convolutional neural network, and the shared layer improves the feature extraction ability of golden crucian carp and obtains better performance and generalization. The architecture of our model is shown in Fig. 1, and the specific experimental procedure and methodology will be pointed out below.

Figure 1 The network structure of our model.

The model contains two tasks, which are the detection task and the segmentation task.

Acquisition of materials

Golden crucian carp is a kind of fish with strong physique, strong resistance and wide eating habits. It is easy to raise and does not require good care. At the same time, the shape of golden crucian carp is similar to crucian carp and grass carp, so the method we proposed can also be well promoted to the cultivation of crucian carp and grass carp in the future. To sum up, we took the golden crucian carp as the experimental object. During the experiment, we collected pictures of fish in a non-intrusive way, without any impact on the fish itself.

We purchased 10 easily distinguishable golden crucian carps in the ornamental fish market and used a transparent tank, 80 cm long, 30 cm wide and 60 cm high, as the breeding environment. The size of the purchased tank and the number of fish is reasonable, so as to ensure that the data collected will not be too concentrated and easy to distinguish. The 10 golden crucian carps were raised in a transparent square tank, provided with sufficient oxygen, and fed with common ornamental fish feed. At the same time, in order to simulate a real environment in the water, we added different doses of water conditioner DEBAO and nitrifying bacteria YUECAI to the initial water environment. Finally, it formed a stable and balanced water environment and achieved the optimal effect.

We used the DJI pocket2 camera to capture golden crucian carps from different angles and distances. In order to ensure the reliability of the experimental data, the images were taken randomly, taking into account the effects of sunlight as well as indoor lighting and other factors. When collecting the dataset, we placed the camera about 15–20 cm away from the front of the fish tank and adjusted the camera angle to ensure that we could get a complete figure of the inside of the fish tank without too much background outside the fish tank, which not only ensure the diversity of the collected images, but also enhance the adaptability of subsequent models to various environments (Lin et al., 2021). In the deep learning model, the quality of the image has a crucial impact on the training results of the model. The higher the quality of the image annotation is, the better the performance of the model may be. Images without golden crucian carp are useless images for data labeling work, and images with golden crucian carp heavily obscured by aquatic plants and images with turbid water will make labeling work difficult. Therefore, for the captured images, we manually screened and removed some images of poor quality (such as images of golden crucian carp not captured, images of golden crucian carp heavily occluded by aquatic plants and images with turbid water).

Finally, we obtained a dataset consisting of 1,858 images from 10 golden crucian carps. There are two categories of datasets, which contain 1,858 general frame object detection datasets and 640 segmentation datasets.

We raised golden crucian carps and photographed them, and our experiments met the ethical requirements for animal welfare, and we obtained the affidavit of approval of animal ethical and welfare approved by the Sichuan Agricultural University IACUC, numbered 20200054. After the data collection was completed, we released the ten golden crucian carps and placed them in an ornamental fish pond on campus, where they could be fed and cared for by relevant professionals.

The above dataset was annotated by 10 annotators under the guidance of professionals. The annotation work is divided into two main parts. In the first part, the object detection dataset was produced. LabelImg is a graphical image annotation tool written in Python language, often used for the production of object detection datasets. The annotations are stored as XML files in PASCAL VOC format, and YOLO format is also supported. Therefore, we used labelImg to frame the target, as shown in Fig. 2, and the annotation is done by framing golden crucian carps’ bodies with a horizontal box to ensure that each fish is completely framed into the corresponding box.

Figure 2 Annotation of object detection dataset.

For the second part, the segmented dataset was created. Labelme, an image annotation tool developed by the Computer Science and Artificial Intelligence Laboratory (CSAIL) of MIT, can be used to annotate images in polygons, polylines, points and other forms, which is a common tool for image annotation in segmentation tasks. Therefore, we used labelme to label golden crucian carp and generate json format for the corresponding image. As shown in Fig. 3, the body outline of each golden crucian carp in the image is framed with dense points, and the number of golden crucian carp in each image varies. Finally, we obtained the annotation results of about 6,000 golden crucian carps, and these datasets were used for the subsequent experiments of semantic segmentation and instance segmentation.

Figure 3 Annotation of the segmentation dataset.

In the subsequent experiments, we divided both the object detection and segmentation datasets into a training dataset, a validation dataset, and a test dataset, all in the ratio of 8:1:1.

Data enhancement for object detection

YOLOV5 passes each batch of training data through the data loader and enhances the training data at the same time. The data loader performs three types of data enhancement: scaling, color space adjustment and mosaic enhancement. For the training of the object detection YOLOv5 model, we used various tricks such as Fliplrud, MixUp, Mosaic, and HSV_Aug for data enhancement on the golden crucian carp dataset, analyzing and comparing the effect of different combinations of tricks on the golden crucian carp dataset, and selecting the most suitable tricks.

Fliplrud

Fish are moving freely in the water and their positions and poses are diverse. The dataset should be contained more information of fish to improve the recognition ability of the model. Therefore, we set up two methods respectively:random horizontal flip and random vertical flip, and the execution probability is 0.5. As shown in Fig. 4, fliplr is flipping the image 180 degrees from left to right or right to left, and flipud is flipping 180 degrees from top down or bottom up.

Figure 4 (A) The picture in our dataset, (B) a picture of (A) flipped left and right, (C) a picture of (A) flipped up and down, and (D) a picture of (A) flipped up, down, left and right.

MixUp

MixUp is an algorithm for mixed-class enhancement of images used in computer vision. It can mix images between different classes, specifically by fusing the features and labels of two samples to expand the training dataset.

We took the fusion ratio λ to obey the beta (α, β) distribution (where the α value range is generally [0.1, 0.4], and the λ value range is between [0, 1]). As shown in Eq. (1), we mix two random samples, and for each input batch image, together with the randomly selected image, it is fused with the randomly selected image, separately for the image itself and the corresponding label according to the fusion ratio λ, to obtain the mixed tensor Mixed_Batch. Among them, the principle of MixUp actually uses linear interpolation.

(1) Mixed_batch=λ∗batch1+(1−λ)∗batch2

Mosaic

Mosaic randomly selects four images from the dataset at a time. After randomly cropping, randomly flipping, and color gamut changes for each image, the four images were placed in the order of top left, top right, bottom left, and bottom right and superimposed to form a new image. As shown in Fig. 5, this is the image obtained after Mosaic operation during the training process. Mosaic not only enriches the background of the dataset images, but also expands the dataset and improves the robustness of the model.

Figure 5 Training images after mosaic operation.

HSV_Aug

RGB is the most common color space that we often come into contact with. The golden crucian carp dataset image we’ve collected is an RGB image. Images acquired in natural environments are often disturbed by factors such as lighting, occlusion, shadows, etc. For such changes in brightness, the three components in the RGB color space are highly correlated with it. In the HSV color space, the change of light can be directly expressed through the change of brightness, which is closer to people’s perception of color and more suitable for image processing. Therefore, we converted the values of the R, G, and B components of the golden crucian carp data into HSV components according to the following Eqs. (2)–(4), thereby obtaining an HSV image. In this way, the characteristics of the image can be expressed more intuitively, and the effect can be significantly enhanced.

(2) V=max(R,G,B)/255

(3) S=(max(R,G,B)−min(R,G,B))/(float)max(R,G,B)/255

(4) H={0∘,ifmax(R,G,B)==min(R,G,B);60∗G−Bmax(R,G,B)−min(R,G,B)+0,ifmax(R,G,B)==R and G≥B;60∗B−Rmax(R,G,B)−min(R,G,B)+120,ifmax(R,G,B)==G;60∗R−Gmax(R,G,B)−min(R,G,B)+240,ifmax(R,G,B)==B;60∗G−Bmax(R,G,B)−min(R,G,B)+360,ifmax(R,G,B)==R and G<B;

FocalLoss

FocalLoss is a kind of dynamic scaling cross-entropy loss based on dichotomous cross-entropy, which focuses on hard-to-distinguish samples by a dynamic scaling factor to reduce the weight of easy-to-distinguish samples in the training process. FocalLoss mainly solves the problem of one-stage object detection cases with extreme imbalance between foreground and background classes during training. The formula is as follows:

(5) FL(pt)=−(1−pt)γlog(pt),

where γ is a constant in the range of values [0, 5], and the formula is as follows:

(6) Pt={pify=11−potherwise,

where the values of y are 1 and −1, representing the foreground and background, respectively, and the values of p are 0~1, representing the probability that the model prediction belongs to the foreground.

By using the above data enhancement techniques, we were able to diversify our dataset, reduce overfitting of the data, and effectively improve the generalization ability of the model.

Methods of detection and estimation

We used the golden crucian carp dataset to test some of the current mainstream instance segmentation networks, and the experimental results are shown in Table 1.

Table 1 Results of the instance segmentation networks.

Model	mAP@0.5	FPS	
Mask R-CNN	0.887	10.6	
YOLACT	0.887	23.0	
Cascade Mask R-CNN	0.916	8.8	

Mean average precision (mAP) is an important measure of the object detection algorithm, which is calculated by the comprehensive weighted average of the average accuracy rate (AP) of all class detection. We calculate the mAP for IoU at 0.5, and the process requires a definite integral to the Precison-Recall curve to find the area. The mAP@0.5 can be calculated as

(7) mAP@0.5=1n∑i=0n−1⁡∫01⁡p(ri)dr,

where n is the number of categories and p(ri) is a Precison-Recall curve function formed by using the model’s Recall as the abscissa and Precision as the ordinate.

FPS refers to the number of images that can be processed by the network model in 1s.

As is seen from Table 1, although these instance segmentation networks have high precision, FPS is low and real-time monitoring is poor. According to the principle of instance segmentation, instance segmentation can be regarded as a combination of object detection and semantic segmentation. Moreover, the current object detection networks (such as Yolov4s, CenterNet, etc.) are fast and have a good real-time detection effect. Therefore, we try to use the combination of object detection and semantic segmentation instead of instance segmentation to achieve real-time detection and segmentation.

In a word, our main methods in this article are as follows: using the Yolov5 network as the backbone network and object detection head, adding semantic segmentation head in Yolov5, using the Ghost module to further optimize the semantic segmentation head, replacing the activation function.

Object detection

In this article, we primarily use the Yolov5 network, version 5.0, as the backbone network and object detection branch, and the full name of YOLO is you only look once, which means that you only need to browse once to identify the category and location of objects in the figure.

The YOLOv5 model was publicly released by Ultralytics on June 9, 2020, and their code and experimental results are open-sourced at https://github.com/ultralytics/yolov5/tree/v5.0. The YOLOv5 model is based on the improved YOLOv3 model, with YOLOv5s, YOLOv5m, YOLOv5l, and YOLOv5x models. In terms of model volume, the number of parameters for YOLOv5s, YOLOv5m, YOLOv5l, and YOLOv5x are 7.3, 21.4, 47.0, and 87.7 M, respectively. The detection speed of YOLOv5s is faster than all the other three models, especially two times faster than YOLOv5x. In general, the smaller the model is, the faster the detection speed is, and it’s also easier to deploy on embedded devices, so we choose YOLOv5s for our subsequent experiments.

YOLOv5 is a One-Stage network, which has improved accuracy and speed compared with other networks before YOLOv5 in the YOLO series, and its network flow is shown in Fig. 6.

Figure 6 The network structure and application of YOLOv5s-5.0.

The CBS component is composed of the Convolutional layer+BatchNormalization+SiLU activation function. The Resunit component draws on the residual structure in the Resnet network and can play a role in building a deeper network. The CSP1_X and CSP2_X component draws on the CSPNet network structure. The FOCUS component is to slice the data, which can play a role in the down-sampling operation without information loss. The SPP component adopts the maximum pooling method of 1 × 1, 5 × 5, 9 × 9, and 13 × 13 for multi-scale fusion.

The network structure of YOLOv5 mainly is composed of Backbone, Neck, and Head, where YOLOv5 uses CSPDarknet as Backbone to extract abundant information features from the input image. Neck utilizes PANet structure, and Neck is mainly used to generate feature pyramids. YOLOv5 uses the same Head as YOLOv3, which is a 1 * 1 convolutional structure with three sets of outputs.

Either Aadm or SGD optimizer can be chosen in YOLOv5, and we use SGD optimizer.

For the CSP structure in YOLOv5, YOLOv5 uses two kinds of CSP structures, the first one mainly uses CSP1_X in Backbone, where the Bottleneck is using the Resunit structure, and the second one is using CSP2_X in Neck, where the Bottleneck does not use the Resunit structure.

YOLOv5 has significant advantages shown as follows: Compared with the Darknet framework used in YOLOv4, using the Pytorch framework is very user-friendly and easy to train your own dataset, and implement into production

It is easy to read code, integration of a large number of computer vision techniques, very conducive to learning and learning

Not only is it easy to configure the environment, but also the model training is rapid, and batch inference can produce real-time results

It is able to effectively inference directly on individual images, batch images, video and even webcam port inputs

The Pytorch weights can be easily converted to ONXX format for Android, which can then be converted to OPENCV format, or to IOS format via CoreML for direct deployment to mobile applications.

The speed of YOLOv5s object recognition is impressively up to 140FPS, and the using experience is very excellent.

Semantic segmentation

In this article, we add a segmentation head to YOLOv5 network for the segmentation task. The segmentation head takes the output of the 16th layer of YOLOv5 as input, and the channel configuration is 512. It is mainly composed of C3, SPP and Up Sampling modules. Our segmentation head network refers to the design idea of Deeplabv3+ (Chen et al., 2018), PSPNet (Zhao et al., 2017) and other networks. The specific network structure is shown in Fig. 7.

Figure 7 Structure of segmentation head.

SPP (He et al., 2015) is the abbreviation of spatial pyramid pooling, which can effectively avoid image distortion caused by image region clipping and scaling operations.

The Up Sampling here uses bilinear upsampling, which is one of the interpolation algorithms and is an extension of linear interpolation. It uses four real existing pixel values around the target point in the original image to jointly determine a pixel value in the target map, and the core idea is to perform linear interpolation once in each of the two directions.

As shown in Fig. 8, we need to ask for the pixel values of point P. We know the coordinates of Q11, Q21, Q12, Q22, and P. The pixel values of Q11, Q21, Q12, and Q22 are also known. So first use the single linear interpolation about X to calculate the pixel values of R1 and R2 respectively, as shown in Eqs. (8) and (9).

Figure 8 Example diagram of bilinear interpolation algorithm.

(8) f(x,y1)≈x2−xx2−x1f(Q11)+x−x1x2−x1f(Q21)

(9) f(x,y2)≈x2−xx2−x1f(Q12)+x−x1x2−x1f(Q22)

The letters f(Q11), f(Q12), f(Q21), f(Q22), x1, x2, and x in the equation on the right are already known, and the derived f(x,y1) and f(x,y2) are the pixel values of R1 and R2.

Then the pixel values of point P are calculated with single linear interpolation about the y direction to obtain Eq. (10).

(10) f(x,y)≈y2−yy2−y1f(x,y1)+y−y1y2−y1f(x,y2)

The letters y1, y2, and y are known in the equation on the right. f(x,y1) and f(x,y2) are the values of R1 and R2 pixels found in the previous equation.

The C3 module in the segmentation head is a feature extraction module designed by YOLOv5 based on the Cross Stage Partial Network (CSPNet) (Wang et al., 2020), in which the stacking of multiple Bottleneck layers causes significant overhead.

In addition, affected by the GhostNet (Han et al., 2020) function of the lightweight network architecture, we replaced the C3 module with the GhostC3 module. The GhostC3 module structure is shown in Fig. 9. The main difference between GhostC3 and C3 is that it replaces the Bottleneck layer in C3 with the GhostBottleneck layer, which makes GhostC3 use fewer parameters, but has higher accuracy than the original. In our test, we found that after replacing C3 module with GhostC3 module, the overall parameters used by the model changed from 7731361B to 7577985B, a decrease of 2%.

Figure 9 The structure of GhostC3 module.

GhostC3 mainly consists of Conv, concat, and GhostBottleneck modules, and the main components of GhostBottleneck module are GhostConv. Among them, the structure of GhostConv is shown in Fig. 10, C1 and C2 are the number of input and output channels, half of the output feature map comes from one regular convolution, and the other half is generated by 5 × 5 Depthwise convolution on the result of the first one. Compared with the original convolutional layer, GhostConv is able to achieve the same or even more efficient feature extraction with less complexity. In CNN networks, it is common to have a lot of convolutional computations with redundant intermediate feature maps, while GhostConvolution forces the network to learn useful features from half of the convolutional kernels.

Figure 10 The structure of the GhostConv module.

The structure of the GhostBottleneck layer is similar to the residual part of the Resnet. Fig. 11 shows the structure for a step size of 1, which consists of two Ghost convolutional stacks with 1 × 1 kernels and residual connections. The GhostBottleneck layer with a step size of 1 can be used to increase the number of channels of the input feature map and to expand the processing for later operations.

Figure 11 The structure of GhostBottleneck(stride = 1).

Figure 12 shows the structure of GhostBottleneck with a step size of 2. Depthwise convolution is applied to downsample the feature map between GhostConv. To ensure that the features in the residual branch have the same dimension as the original features, the Depthwise convolution is used for downsampling and the original 1 × 1 convolution is used to change the number of channels. The GhostBottleneck layer with a step size of 2 enables feature extraction by stacked GhostConv when downsampling the feature map and lifting the channel. At the same time, it mitigates the gradient disappearance problem by residual concatenation.

Figure 12 The structure of GhostBottleneck(stride = 2).

The GhostBottleneck structure approach can not only effectively reduce the model parameters and computation, but also improves the detection efficiency of the model by optimizing the feature map with the Ghost module. Therefore, we replaced the original C3 module with the GhostC3 module.

Activation function improvement

The YOLOv5 model provides a variety of activation functions, such as ReLU, Swish, etc. The activation functions can increase the nonlinear factors. The linear model is not sufficiently expressive, and a nonlinear function is introduced as the excitation function so that the deep neural network is more expressive.

Therefore, to further improve the accuracy of our model, for the activation function, we replaced the Swish activation function used Conv of the YOLOv5 model with the HardSwish activation function.

The Swish activation function (as shown in Fig. 13) replaces the ReLU activation function, which can significantly improve the accuracy of the neural network, and the specific definition of the Swish activation function is shown in Eq. (11).

Figure 13 Swish function diagram.

(11) f(x)=x∗sigmoid(x)

Although this non-linearity improves the precision, the sigmoid function is composed of exponentials, which are much more computationally expensive on mobile devices. The sigmoid activation function can be fitted with the segmented linear function HardSigmoid, as shown in Eq. (12).

(12) Hardsigmoid(x)={ 0,    x≤-3 1,    x≥3x6+12   otherwise

As a consequence, replacing sigmoid with Hardsigmoid can greatly reduce the cost of operations, which led to the birth of HardSwish (As shown in Fig. 14), as specified in the Eq. (13).

Figure 14 HardSwish function diagram.

(13) HardSwish(x)=x*Hardsigmoid(x)=x*ReLU6(x+3)6=x*{ 1,     x≥3x6+12,   −3<x<3 0,     x≤-3

The derivative of this function with respect to x is:

(14) HardSwish′(x)={1,x≥3x3+12,−3<x<30,x≤−3

Results

Our experiments are divided into three main steps.

First, we validate multiple object detection models, and then select the network with better performance as the baseline to be used as the model in the first step.

Second, we add a segmentation head to the baseline model and perform ablation experiment on it to select the most suitable tricks.

Third, we further optimize the proposed model and compare the results with those produced by other models to demonstrate the competitive advantage of our approach.

Through stepwise experiments, we can ensure that each step of our identification of golden crucian carp is a local optimal solution, thus an effective model is built for the detection and segmentation of golden crucian carp.

Model selection

Object detection model selection

Currently, the mainstream target detection models are CenterNet, YOLOv4, YOLOv5, EfficientDet, and RatinaNet, and we trained these five target detection models on the existing golden crucian carp dataset, and the results of these model tests are listed in Table 2. To ensure the accuracy of comparison between the results of each model above, we uniformly conducted the experiments on RTX 5000 with uniform settings of parameters: image size of 640 × 640, optimizer of Adamw, initial learning rate set to 0.001, and epoch of 300.

Table 2 Comparison of object detection models.

Model	Precision	Recall	F1-score	mAP@0.5	mAP@0.5:0.95	Inference
@batch_size 1 (ms)	
CenterNet	95.21%	92.48%	0.94	94.96%	56.38%	32	
Yolov4s	84.24%	94.42%	0.89	95.28%	52.75%	10	
Yolov5s	92.39%	95.38%	0.94	95.38%	58.31%	8	
EfficientDet	88.14%	91.91%	0.90	95.19%	53.43%	128	
RatinaNet	88.16%	93.21%	0.91	96.16%	57.29%	48	
Note:

Refers from Lin et al. (2021).

In the above experiments, we choose Precison, Recall, F1-score, mAP@0.5, mAP@0.5:0.95 and Inference@batch_size 1 as the performance evaluation indexes for model effectiveness, and we consider these indexes together to determine the selection of our final model. Through Table 2, we find that CenterNet has the highest precision of 95.21% on the golden crucian carp dataset, but its Recall value is lower compared to other networks, only reaching 92.48%, while YOLOv5s Recall reaches 95.38%. Therefore, we reconcile Precision and Recall for the mean. And we summed Precision and Recall to obtain the F1-score, which is exactly the same. But in terms of mAP, YOLOv5s is better. At the same time, from the overall point of view, YOLOv5s reaches the top of each index compared with the rest of the network models, so we finally choose YOLOv5s.

In the experimental prediction process, the following four cases occur for the golden crucian carp dataset: (a) correctly identify the golden crucian carp, indicated by TP (b) incorrectly identified other things as golden crucian carp, denoted by FP (c) did not identify golden crucian carp, denoted by FN (d) non-golden crucian carp things were not identified as golden crucian carp, denoted by TN. To obtain the specific values of these three cases, the calculation of IoU is required, which is the degree of overlap between the predicted boxes and the boxes marked in the original figure, as shown in Fig. 15, which is calculated as in Eq. (15) (where G is the real box and D is the predicted box), and in general, we write down the number of detected boxes with IoU greater than 0.5 as TP, while the data of detected boxes whose values are less than or equal to 0.5 are written down as FP.

Figure 15 Visual representation of IoU.

(15) IoU=G∩DG∪D

Using TP, FP and FN, we can calculate the above four metrics in the following way.

As shown in Eq. (16), Precison is the ratio of the number of correct predictions in the forecast results.

(16) Precison=TPTP+FP

Recall denotes the probability of being detected in all positive samples during the prediction of experimental results, which in this experiment is the probability of being detected in all pictures containing golden crucian carp, and is calculated as in Eq. (17).

(17) Recall=TPTP+FN

The F1-score can well distinguish the strengths and weaknesses of the algorithm and is the summed average of Precision and Recall, as shown in Eq. (18).

(18) F1−score=2×Recall×PrecisonRecall+Precison

mAP@0.5:0.95 represents the average mAP at different IoU thresholds (from 0.5 to 0.95 in steps 0.05).

Inference@batch_size 1 indicates the inference time required for a picture.

The experimental results show that YOLOv5s and CenterNet show relatively better performance, with F1-scores reaching 0.94 and mAP@0.5 around 0.95. However, the Inference@batch_size 1 of YOLOv5 is smaller than CenterNet, and takes less inference time to process the same photo, so we chose YOLOv5s as the backbone network for subsequent experiments.

Selection of data enhancement tricks

We added a semantic segmentation head in YOLOv5 for subsequent experiments. To further improve the effectiveness of our model and enhance its generalization ability, we used different tricks to process the model and further ablation experiments were conducted to extract the best model training configuration. Tables 3 and 4 show the effects of using tricks such as HSV_Aug, FocalLoss, Mosaic, MixUp and the evaluated metrics afterwards. This symbol,“√” indicates that the technique was used and “×” indicates that it was not used.

Table 3 YOLOv5s+segmentation head with different data enhancement tricks.

HSV_Aug	FocalLoss	Mosaic	MixUp	Fliplrud	Group	
×	×	×	×	×	1	
√	×	×	×	×	2	
√	√	×	×	×	3	
√	√	√	×	×	4	
√	√	√	√	×	5	
√	√	×	×	√	6	
√	√	√	×	√	7	
√	×	√	×	×	8	
√	×	√	√	×	9	
√	×	√	×	√	10	
√	×	√	√	√	11	
√	√	√	√	√	12	

Table 4 The experiment results of different data enhancement tricks.

Group	Precision	Recall	F1-score	mAP@0.5	Seg Acc	Seg mIoU	FPS	
1	0.904	0.931	0.917	0.955	0.981	0.978	117.1	
2	0.905	0.935	0.920	0.963	0.980	0.977	115.0	
3	0.888	0.924	0.906	0.947	0.975	0.972	114.4	
4	0.922	0.939	0.930	0.966	0.985	0.983	114.8	
5	0.904	0.924	0.914	0.957	0.984	0.982	113.8	
6	0.897	0.918	0.907	0.942	0.985	0.983	118.7	
7	0.899	0.937	0.918	0.955	0.982	0.980	113.4	
8	0.954	0.917	0.935	0.976	0.985	0.984	115.1	
9	0.952	0.924	0.938	0.974	0.986	0.984	116.2	
10	0.935	0.924	0.929	0.964	0.984	0.982	115.3	
11	0.893	0.910	0.901	0.943	0.986	0.984	114.3	
12	0.900	0.936	0.918	0.954	0.986	0.984	113.3	

Considering the connection and impact in different data enhancement strategies (e.g., MixUp (Zhang et al., 2017) is implemented on top of Mosaic, thus MixUp is only used as the Mosaic strategy is enabled), we finally selected only 12 combinations as shown in Table 3 for our experiments.

The results of the experiments are shown in Table 4, and the results show that some tricks decrease the accuracy of our network, while some tricks increase the accuracy of the network.

For the semantic segmentation model, mIoU is its standard metric. The formula of mIoU is shown in Eq. (19), which is labeled as Seg mIoU in Table 4. In our experiment, the golden crucian carp dataset is mainly divided into two categories, fish and background, so k is 1.

(19) mIoU=1k+1∑i=0kTPFN+FP+TP

seg ACC refers to the pixel accuracy, and its corresponding formula is shown in Eq. (20).

(20) segACC=(TP+TN)/(TP+TN+FP+FN)

FocalLoss is often used to suppress background classes in unbalanced data and target detection (Li et al., 2022). As can be seen in the results of Table 4 for both Group 2 and 3, and Group 7 and 10, the accuracy of the model after FocalLoss processing decreases with the golden crucian carp dataset and the corresponding settings in this article. We speculate that there is no significant imbalance in the space allocation between background and object classes in the data, causing FocalLoss not to work.

All of our detectors were experimented using pre-trained weights. The experimental results of Groups 3 and 4 can clearly see that Mosaic has a certain effect on improving the precision of the model, and the experimental results of Groups 6 and 7 can also show that Mosaic can improve the performance of the model, so we decided to use Mosaic. Looking at the experiments in Groups 8 and 9, we can see that MixUp has little effect on model precision, recall and other parameters. Group 5 added MixUp to Group 4, and the precision of the model decreased, and Group 11 added MixUp to Group 10, and the performance of the model also decreased, so we decided not to use MixUp. Similarly, the experiments in Groups 7 and 8 led us to reject Fliplrud. Therefore, we chose the tricks of Group 8, that is, we only used HSV_Aug and Mosaic for subsequent experiments.

Parameter setting

In the Yolov5s model with segmentation head, we finally used the 8th set of tricks as in Table 4, and we set the hyperparameters of HSV color model as H (hue): 0.015, S (saturation): 0.7, V (brightness): 0.4, and mosaic as 1. We set the image input size as 640 * 640, epochs We set the input size to 640 * 640, epochs to 300, SGD optimizer and initial learning rate to 0.0015, and put the custom golden crucian carp dataset with batch size of 2 into the network for training, and our experimental environment for training is ubuntu 20.04 with RTX3060 graphics card.

Experimental results

Our final model uses YOLOv5s version 5.0 as the backbone network, and adds a segmentation head with Ghost module optimization, and replaces the original Swish activation function with the HardSwish activation function, which is relatively less computationally expensive. The final object detection accuracy is 0.954, the recall is 0.930, and the speed is 116.6 FPS on RTX3060.

The experimental results after improving the semantic segmentation head and replacing the activation function are shown in Table 5.

Table 5 Experimental results after improving the semantic segmentation head and replacing the activation function.

YOLOv5s+segmentation head(C3) is Group 8 of Table 4, YOLOv5s+segmentation head(GhostC3) means replace C3 in the segmentation head with GhostC3, YOLOv5s +segmentation head (GhostC3)+HardSwish means replacing C3 in the segmentation head with GhostC3 and replacing the original Swish activation function with the HardSwish activation function.

Group	Precision	Recall	F1-score	mAP@0.5	Seg Acc	Seg mIoU	FPS	
YOLOv5s+segmentation head(C3)	0.954	0.917	0.935	0.976	0.985	0.984	115.1	
YOLOv5s+segmentation head(GhostC3)	0.952	0.927	0.939	0.975	0.985	0.983	116.0	
YOLOv5s+segmentation head(GhostC3)+
HardSwish	0.954	0.930	0.942	0.976	0.985	0.983	116.6	

According to Eq. (16), we can see that precision refers to the probability of True positive in all samples that are predicted to be correct, mainly focusing on “finding precision”; According to Eq. (17), we can see that recall refers to the probability that True positive accounts for the sample that is actually True, and focuses mainly on “finding the full”. If you just look at recall or precision, you may go to extremes without knowing it. We expect the results to be both accurate and complete, so we choose F1-score, which combines precision and recall, as the evaluation metric. We tend to choose the model with a higher F1-score.

After replacing the C3 module in segmentation head with the GhostC3 module, the F1-score and FPS values of the model were improved compared with the results of the initial version of the segmentation head model. After replacing the Swish activation function with the HardSwish activation function on the segmentation head model improved with the Ghost module, the F1-score of the model increase to more than 0.94 and the FPS increase by 0.6, which is relatively obvious. Therefore, we will use the model of the Ghost module that has optimized the segmentation head and the activation function is HardSwish as our final model.

As shown in Table 6, our multi-task model shows advantages over other models in both detection and segmentation tasks. In particular, in terms of speed, the FPS of our model is much higher than the other models. To further demonstrate the effectiveness of our model, we conduct experiments on the publicly available dataset PASCAL VOC 2007. We divide the dataset into training set, validation set and test set in the ratio of 8:1:1, and the size of the training images is set to 512 * 512. Finally, our object detection accuracy is 0.738, semantic segmentation accuracy is 0.843, and FPS is 120. We use our final model to make predictions for the golden crucian carp videos and take four images from it, as shown in Fig. 16.

Table 6 Comparison results of different models in segmentation task and detection task.

Model	mAP@0.5	Seg Acc	Seg mIoU	FPS	
FCN	\	0.982	0.965	8.6	
OCRNet	\	0.985	0.967	10.1	
Deeplabv3+	\	0.981	0.966	8.0	
UPerNet	\	0.982	0.966	8.6	
ANN	\	0.982	0.965	7.0	
CcNet	\	0.982	0.965	6.9	
DANet	\	0.982	0.966	8.0	
PSPNet	\	0.981	0.965	9.1	
Mask R-CNN	0.887	\	\	10.6	
YOLACT	0.887	\	\	23.0	
Cascade Mask R-CNN	0.916	\	\	8.8	
YOLOv5s+segmentation head(GhostC3)+HardSwish
(Our method)	0.976	0.985	0.983	116.6	

Figure 16 The results of our model for detecting and segmenting the golden crucian carp.

From this prediction result, we can see the video prediction effect of our model: high accuracy of object detection and high coincidence of semantic segmentation with the object to be detected.

Discussion

The current development trend shows that instance segmentation technology has a large potential for application in smart fisheries. However, the problems of instance segmentation, such as low speed and inferior real-time detection, have contributed to the poor results of its application. In this article, we study a multi-task model based on YOLOv5, add a segmentation head for semantic segmentation based on the original YOLOv5 framework, improve the segmentation head by the Ghost module, and replace the activation function of the model to further improve the overall accuracy of the model. Based on this, our discussions are as follows.

Contribution to smart fisheries

Fishery is an indispensable part of China’s national economy. With the continuous improvement of people’s living standards, the process of fish farming has received wide social attention. At all the stages of fish growth and development are susceptible to a variety of external factors, possibly resulting in poor fish growth or even death, thus causing serious losses to farmers. Therefore, the major focus of current research on smart fisheries is how to analyze and understand the state of fish growth process in a timely and effective manner. To address this problem, researchers have proposed many effective methods. Among them, real-time monitoring of fish using deep learning techniques is by far the most effective method. In this article, we add the optimized segmentation head of Ghost module to the YOLOv5s object detection model to achieve the function of real-time detection and segmentation, and experimentally prove that our method greatly improves the detection speed in real-time monitoring, and the object detection accuracy and semantic segmentation accuracy are above 0.95. Therefore, our method can be applied to the smart fishery aspect, which is important for the monitoring activities of fish growth and development process.

Limitations and future work

At present, our model mainly utilizes a object detection algorithm with a semantic segmentation algorithm, and the segmentation head technique is used to effectively combine the two algorithms to form a new network model that guarantees the detection of the object while being able to segment it. Our experiments test on the existing golden crucian carp dataset, and the final results have achieved an accuracy of more than 95% and an FPS of 116.6, showing excellent results. However, our experiments still have some limitations. Our dataset is only for the golden crucian carp, and we have not yet experimented the effect of the model on the dataset of common fish such as grass carp and crucian carp. In the future, we will work on the expansion of the original golden crucian carp dataset to diversify the fish species to ensure that the dataset can contain more fish with generalizability, and we will also keep on expanding the quantities of each type of fish to form a large-scale fish dataset. On this foundation, we will conduct further experiments and optimization on the basis of this model to ensure the universality of the model. Our ultimately purpose is to implement the model in practice, and we intend to develop it into a system that can estimate the biomass of fish and continuously optimize and expand its functions to effectively promote the intelligent development of fishery farming.

Conclusions

In this article, we first detected and segmented golden crucian carps by the general instance segmentation algorithm. Although the experimental results were comparatively good in terms of accuracy, each FPS was low and it was difficult to reach the standard of real-time monitoring. Subsequently, we added the Ghost module optimized semantic segmentation head to the YOLOv5 model and replaced the activation function, and finally propose our model. It is experimentally confirmed that our model can effectively identify grass carp and perform real-time segmentation, with object detection accuracy up to 95.4%, semantic segmentation accuracy up to 98.5%, and a speed of 116.6 FPS on RTX3060, which can effectively meet the requirements related to real-time detection.

In the future, our research will be further developed in the following aspects.

1. In this article, the survival environment of our fish is a fish tank, but in the actual farm, it may be more complex. For this reason, we will consider further optimization of the experimental environment to ensure a closer fit to the farm’s culture environment and increase the robustness of the model.

2. Grass carp is a unique fish species and an important freshwater economic fish in China. Considering its mature farming technology, low-cost input, little difficulty in management, high survival rate and high economic value, we choose grass carp as the research target. However, the characteristics of grass carp are not obvious enough compared with golden crucian carp. For this reason, we intend to add some hardware such as one-way pipes to assist us in collecting more detailed and comprehensive data sampling of grass carp to avoid the problem of poor model training results caused by insufficient features.

3. We will further extend our features to perform fish pose estimation, behavior recognition, target detection of fish diseases and semantic segmentation of fish diseases based on semantic segmentation, in order to form a completely intelligent fish farming system and improve the foundation for future application in actual farms.

Supplemental Information

Supplemental Information 1 Third party source code.

Click here for additional data file.

Supplemental Information 2 Our improved YOLOv5 code.

We improve on YOLOv5 so that it can perform both object detection and semantic segmentation.

Click here for additional data file.

Supplemental Information 3 The source where we obtained the third-party code used in our analysis.

Click here for additional data file.

Supplemental Information 4 ARRIVE 2.0 Checklist.

Refers from Lin et al. (2021).

Click here for additional data file.

We are grateful to Prof. Duan and other teachers for their support and assistance. It was with their help that we could carry out research so smoothly during our college life.

Additional Information and Declarations

Competing Interests

Author Contributions

Ethics

Data Availability

The authors declare that they have no competing interests.

QinLi Liu conceived and designed the experiments, performed the experiments, analyzed the data, prepared figures and/or tables, authored or reviewed drafts of the article, and approved the final draft.

Xinyao Gong conceived and designed the experiments, performed the experiments, prepared figures and/or tables, and approved the final draft.

Jiao Li performed the experiments, authored or reviewed drafts of the article, and approved the final draft.

Hongjie Wang performed the experiments, performed the computation work, prepared figures and/or tables, and approved the final draft.

Ran Liu performed the experiments, prepared figures and/or tables, and approved the final draft.

Dan Liu performed the experiments, prepared figures and/or tables, and approved the final draft.

Ruoran Zhou analyzed the data, prepared figures and/or tables, and approved the final draft.

Tianyu Xie performed the computation work, authored or reviewed drafts of the article, and approved the final draft.

Ruijie Fu performed the computation work, authored or reviewed drafts of the article, and approved the final draft.

Xuliang Duan conceived and designed the experiments, authored or reviewed drafts of the article, and approved the final draft.

The following information was supplied relating to ethical approvals (i.e., approving body and any reference numbers):

Sichuan Agricultural University IACUC.

The following information was supplied regarding data availability:

The raw data and code are available in the Supplemental Files.

Our improved YOLOv5 code is available at Zenodo:

qinli liu. (2022). Our code for realtime fish detection and segmentation. Zenodo. https://doi.org/10.5281/zenodo.7413306.

The detection dataset used for modeling is available at Zenodo:

qinli liu. (2022). Golden crucian carp detection dataset [Data set]. Zenodo. https://doi.org/10.5281/zenodo.7413354.

The segmentation dataset used for modeling is available at Zenodo:

qinli liu. (2022). Golden crucian carp segmentation dataset [Data set]. Zenodo. https://doi.org/10.5281/zenodo.7410176.

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
