# Peer review of "A multitask model for realtime fish detection and segmentation based on YOLOv5"

_PeerJ Computer Science, doi:10.7717/peerj-cs.1262_

## Round 0.1 · original submission · Major Revisions

Revise as per the reviewers’ comments.

Reviewer 1 ·

Basic reporting

The authors have argued for a learning problem and narrated a solution to it. The article is mostly well-written, builds on SOTA, and represents a contribution to the literature.
In terms of basic reporting, it is suggested:
1. Re-write the manuscript for conciseness so that more readers could gain from the work
2. To check carefully for errors; To cite two examples:
(i) Equation 10. LHS should be HardSwish
(ii) Equtions 4: final characters missing
3. To check for typos and grammar.
4. Some literature references are incomplete. For e.g,
Redmon J , Farhadi A . YOLOv3: An Incremental Improvement[J]. arXiv e-prints, 2018.
(arxiv ID to be provided)

Experimental design

The research question is meaningful and has been addressed rigorously using hybrid deep learning architectures. The authors could provide more diagrams or explanations for their precise innovation to the existing models that they have adapted, and the effect of the innovations they have introduced.

Validity of the findings

(1) The findings are valid, and the authors' model appears to Recall well. But their model does not score well on the precision metric, and authors could discuss this situation in detail.
(2) Secondly, the authors may provide the original Source image dataset that has been collected specifically for this study. Replication of the study is not possible without the source dataset, and the availability of high-quality open-source datasets would spur the research in this area of multi-instance segmentation.

Reviewer 2 ·

Basic reporting

1) The English expressions in this paper are not professional enough.
2) The original data were not shared as supplementary files.

Experimental design

1) In this paper, YOLOv5 combined with semantic segmentation head is used for real-time fish detection and segmentation. This method has already been applied in many related researches and lacks of innovation.
2) In the Section of Introduction, the authors stated that “Since there is a relationship between the weight of organisms and their body length and image area, the weight can be estimated indirectly through a deep learning-based image segmentation method …”. Since the image area of fish depends not only on the size of the organism but also on the distance between the camera and the organism, it seems difficult to estimate the weight from the photos taken by camera.

Validity of the findings

No comment.

Additional comments

No comment.

Reviewer 3 ·

Basic reporting

The whole manuscript is written in a very confusing and unclear way. Many paragraphs, in particular in the materials and methods, are very difficult to understand. The manuscript requires a thorough review before being accepted.
The introduction addresses the relevant questions to be answered. The text is however not very scientific and could be clearly improved with a better linkage between the ideas.
The cited literature is adequate. It could be enhanced with a broader review.
The overall structure conforms, but the content is marginally relevant and is poorly described.
The golden crucian carp dataset is not clearly identified in the shared datasets.
The manuscript is self contained but results are poorly presented and discussed. Hypotheses are not conveyed.

Experimental design

Research is in the scope of the journal.
The research question are reasonably defined and are important. The authors state that the research fills specific gaps but the results and discussion sections do not convey the adequate information.
It is very difficult to analyse the research that was performed because the manuscript has a very confusing way to present methodology and results.
The methods are presented in a very unclear way and there are several flaws on figure citing and analysis. Methods needs further referencing and further presentation.

Validity of the findings

The impact and novelty of the research is hard to assess given that the dataset is limited and is not in a real aquaculture settings (where currents and sediments may play an important role).
The carp dataset is not clearly identified in the raw data sets provided.
Conclusions need a deep review to identify clearly the contributions and the questions to be addressed in the future.

Additional comments

Other comments:
Abstract:
The reviewer believes that the sentence “most of the instance segmentation networks have a general efectiveness in real-time monitoring” conveys a positive message and the goal of the authors was the opposite one. Please review the sentence for clarity.
The abstract is missing a conclusion of general applicability on the proposed method. While good results for a specific set are important, the relocatable nature of the methodology should be included.
Introduction
Improve of English:
• (lines 29-31): sentence “In the process of the agriculture, managers need to gain information about the life of fish, shrimp, shellfish and other aquatic organisms, specifically, species, behavior identification and biomass estimation. Among them, the biomass estimation is the total weight of fish and shrimps in a specific water.” Should be substituted by: “In the process of the agriculture, managers need to gain information about the life of fish, shrimp, shellfish and other aquatic organisms, specifically, species, behavior identification and biomass estimation (total weight of fish and shrimps in a specific water).”
• Line 43: remove word “Meanwhile”
Line 48: The 4 tasks should be described and adequate references identified for each (or all ) of them.
Line 50 – are there other object detection algorithms besides those based in deep learning. If yes, review them also.
The whole chapter would benefit from a review to improve the English and the sequence of ideas. Some sentences seem to have little relation with the previous ones (e.g. lines 55-58, 72-73).
Line 114 – it is unclear what “New attempts” mean in the context of the list
Materials & Methods
The overall methodology is confusing and a clear workflow with inputs and outputs at each stage is missing. Examples of application of the enhancers are also necessary.
Details:
Lines 124-132 – review for clarity
Line 144 – when the authors refer “light”, are you referring to artificial light?
Line 147-149 – these situations occur in real life and the data extraction should be able to handle them. We invited the authors to justify the arguments for this removal and to explain whether a manual or automatic procedure was used.
Section 2.2.3 is unclear. Please improve the clarity of the text.
Sections 2.2.1 – explain the need for Fliplrud and what does this method do, for the readers that are not experts
Line 192 “Figure 3: As in figure (b) (c) (d) ” which figure?
Line 222 – define what is FocalLoss and provide a reference/site for it.
Lines 226-230 – very unclear text, rephrase in a clear way.
Line 243: Figure 4: YOLOv5s-5.0 model diagram. – This figure is an image not a diagram
Figure 5 and 6 are also inconsistent between the text and the list of figures.
Figure 7: too small and with bad quality. No labels are provided, so it is unclear what each box refers too. The same problem applies to figure 8. As a consequence the corresponding text is unclear (lines 272-304).

Line 297 – provide a reference for Ghost C3 module and the differences between it and C3.
Results
368 3.1.1. Object detection model selection – this section is completely nuclear. The values of the evaluation are based on the application to the carp dataset? Under what conditions?
Lines 378 and following – the series of application of tools does not follow what is described in the materials and methods. It is quite unclear and not linked to figures. This whole section needs a deep review for content, clarity and organization.
The proposed method(s) should be clearly identified in each table.
Line 436 “3.1.3. YOLOv5 ablation experiment”- this section needs to be clarified as the generic view is not conveyed to the reader.
Several indicators need definition (e.g. Seg Acc)
This section should convey the principal results of the proposed method(s) in particular in comparison with previously existing methods. The principal motivation of speed and usage in real time mode is not clearly identified in the proposed manuscript. Therefore, the “results” part needs to be revised in detail to address the above comments and to comply with the goals of the manuscript.
Discussion
Lines 510 to 519 belong in the introduction. A summary could be mentioned here but just in the context of how the proposed methods go beyond this SoA.
Conclusions
The carp dataset is here identified as one of the contributions of the paper, but that is not consistent with the remaining of the paper. The characteristic of the dataset does not appear to be innovative. The rationale in the introduction is much more clear, by identifying that this dataset had to be created because others were not available.
The description of the contributions is very difficult to follow as it mixes the general approach with very fine details. Please revise.
A paragraph on what were the contributions of the paper and how they can be used in a generic way is missing.

---

## Round 0.2 · Minor Revisions

Revise as per reviewer comments.

Reviewer 1 ·

Basic reporting

The authors have addressed the comments raised by this reviewer in the first round, and the reporting is certainly much improved, and the manuscript readable.
In l180: photographed from different angles and distances:
--> this can be explained better
l201: labelImg
--> some context for this tool
l205: labelme
--> some context for this tool

Eqn(5) --> variables not explained
Eqn(6) --> same

l344: misleading to give the numbers, better to state the % reduction

Metrics such mAP may also need to be explained, for an international audience.

Experimental design

l293-295: YOLOv5 -- four models, and we use the smallest and the most fundamental model
--> Why not use the best model? This can be better motivated.

l456: There are 32 possibiliies for exploring the space of feature combinations, and the authors have only explored twelve different combinations. Hence the statement is contradictory.

Validity of the findings

The findings, subject to the above suggestions, seem valid.

Reviewer 2 ·

Basic reporting

no comment

Experimental design

no comment

Validity of the findings

no comment

Additional comments

The comments have been well addressed.

---

## Round 0.3 · accepted · Accept

This article can be accepted now.